# Effects of Austenitizing Temperature on Tensile and Impact Properties of a Martensitic Stainless Steel Containing Metastable Retained Austenite

**DOI:** 10.3390/ma14041000

**Published:** 2021-02-20

**Authors:** Biao Deng, Dapeng Yang, Guodong Wang, Ziyong Hou, Hongliang Yi

**Affiliations:** 1The State Key Laboratory of Rolling and Automation, Northeastern University, Shenyang 110819, China; dengbwel@163.com (B.D.); yangdapeng6@163.com (D.Y.); wanggd@mail.neu.edu.cn (G.W.); 2Department of Materials Science and Engineering, KTH Royal Institute of Technology, SE-100 44 Stockholm, Sweden

**Keywords:** austenitizing temperature, martensitic stainless steels, retained austenite, ductility, impact toughness, carbides

## Abstract

Austenitizing temperature is one decisive factor for the mechanical properties of medium carbon martensitic stainless steels (MCMSSs). In the present work, the effects of austenitizing temperature (1000, 1020, 1040 and 1060 °C) on the microstructure and mechanical properties of MCMSSs containing metastable retained austenite (RA) were investigated by means of electron microscopy, X-ray diffraction (XRD), as well as tensile and impact toughness tests. Results suggest that the microstructure including an area fraction of undissolved M_23_C_6_, carbon and chromium content in matrix, prior austenite grain size (PAGS), fraction and composition of RA in studied MCMSSs varies with employed austenitizing temperature. By optimizing austenitizing temperature (1060 °C for 40 min) and tempering (250 °C for 30 min) heat treatments, the MCMSS demonstrates excellent mechanical properties with the ultimate tensile strength of 1740 ± 8 MPa, a yield strength of 1237 ± 19 MPa, total elongation (ductility) of 10.3 ± 0.7% and impact toughness of 94.6 ± 8.0 Jcm^−2^ at room temperature. The increased ductility of alloys is mainly attributed to the RA with a suitable stability via a transformation-induced plasticity (TRIP) effect, and a matrix containing reduced carbon and chromium content. However, the impact toughness of MCMSSs largely depends on M_23_C_6_ carbides.

## 1. Introduction

Medium carbon martensitic stainless steels (MCMSSs) are commonly used for manufacturing components, such as cutlery, dental and surgical instruments, valve trim and parts, shafts, and plastic molding [1,2], due to a good combination of high strength and medium corrosion resistance. A typical heat-treatment process for MCMSSs after hot rolling consists of austenitizing, quenching and tempering [2,3], during which the parameters, e.g., austenitizing temperature and time or the quenching finish temperature, of each process affect its mechanical properties [2,4]. Among the aforementioned parameters, the austenitizing temperature plays a decisive role in the initial microstructure, including the fraction of undissolved Cr-rich carbides and austenite in matrix, which will be inherited in the final products in the following process [5]. Clearly, the austenitizing temperature exerts an important influence on the mechanical properties of MCMSSs [1,2,5,6,7,8]. Therefore, understanding the effects of austenitizing temperature is crucial for MCMSS production.

The effects of austenitizing temperature on the microstructure and mechanical properties of martensitic stainless steels, being an attractive topic within academia and the industrial community, have been investigated widely, together with the quenching and tempering (Q-T) process [2,5,6]. Barlow [2] studied the effects of austenitizing temperature on the austenite grain size, hardness, retained austenite (RA) volume faction and the carbides density of AISI 420 stainless steels systematically. Rajasekhar et al. [6] reported that the ductility and impact toughness of AISI 431 stainless steels, which were subjected to double tempering, firstly increased and then decreased with the austenitizing temperature increasing from 950 to 1150 °C. In particular, retained austenite (RA) has been recently introduced to MCMSSs [9,10,11,12], by e.g., quenching and partitioning process (Q&P), aiming to further increase ductility and toughness simultaneously. Concerning the content of RA, the martensite starting (*M*_s_) and finishing temperature (*M*_f_), which depend on alloying elements and austenitizing temperature [13,14], are key parameters to determine the fraction of RA at room temperature. The tempering temperature also affects the content of RA due to the partitioning of alloying elements and phase decomposition [15], but this is not the topic here. Koistinen and Marburger suggested that the content of RA is increased when the *M*_s_ temperature decreases [16]. However, to the authors’ best knowledge, the study of austenitizing temperature on the microstructure and mechanical properties of MCMSSs containing metastable RA is rare. 

In the present work, the influence of austenitizing temperature has been studied on the microstructure and mechanical properties of an MCMSS. Here, the austenitizing temperature is the unique variable parameter to adjust Cr-rich carbides including the fraction and size in the studied alloy. To reduce the sensibility of the initial fraction of austenite to quenching temperature, the quenching temperature is designed as an ambient temperature by a combination of alloy design. By revealing the microstructure and mechanical properties of MCMSSs along with comparisons to different austenitizing temperatures from experimental data, the intrinsic link among process-microstructure-properties has been discussed in detail, aiming to distinguish which is the decisive factor and how it improves the ductility and impact toughness of the MCMSSs.

## 2. Experimental Procedure

### 2.1. Material Preparation

The chemical composition of the studied alloy was Fe-0.23C-2.3Mn-1.5Si-12.5Cr–0.03Ti-0.05Nb wt.% [12]. The alloy was manufactured as a 50 kg ingot by vacuum induction furnace. The cast ingot was forged into slabs with dimensions of 300 mm× 60 mm × 30 mm at 1200 °C and air cooled to room temperature. Subsequently, the slabs were homogenized at 1200 °C for 40 min and hot-rolled to 7 mm in thickness, followed by air cooling to room temperature. Prior to the heat treatment, the hot-rolled plates were subjected to softening at 700 °C for 2 h followed by air cooling. Then, the plates with thickness of 5.5 mm were: a) austenitized at 1000/1020/1040/1060 °C for 40 min, followed by oil quenching to room temperature; b) tempered for 30 min at 250 °C for carbon partitioning followed by air cooling to room temperature. The samples subjected to a) were named 1000-Q, 1020-Q, 1040-Q and 1060-Q, respectively, and the samples subjected to a) and b) were named 1000-P, 1020-P, 1040-P and 1060-P, respectively. In addition, to limit the variable factors in microstructure, a comparison sample, which has an identical process to 1060-P, but directly quenching to liquid N_2_ (−196 °C) other than to room temperature, was designed, and named sample 1060-NP. Through this, the initial microstructure before quenching in sample 1060-NP is the same as that in sample 1060-P. Thus, the key microstructure of RA in terms of both ductility and impact toughness can be discussed and determined. 

The austenite transformation starting and finishing (*A*_c1_ and *A*_c3_, respectively) temperatures during heating were measured by a dilatometer (Bähr DIL 805A/D, TA Instruments, Newcastle, DE, USA) using cylindrical dilatometric samples of 10 mm length and 4 mm diameter under a high purify nitrogen atmosphere, where the heating rate was 10 °C/s. The *M*_s_ temperature was measured by same way but with a cooling rate 30 °C/s. RA determinations were carried out on a D/max-2400 X-ray diffractometer (Rigaku, Tokyo, Japan) (XRD) using Cu radiation. The samples were scanned over a 2θ range from 40° to 100° with angular steps of 0.04° and a scan speed of 2° min^−1^. The volume fraction of RA was evaluated from diffraction peaks of α(200), α(211), γ(200), γ(220) and γ(311) following the equation [17]:(1)$$V_{\gamma }=1{.4I}_{\gamma }/(1{.4I}_{\gamma }{+I}_{\alpha })$$
where the $V_{\gamma }$ is the volume fraction of RA, $I_{\gamma }$ is the average integrated intensity of austenite peaks, and $I_{\alpha }\;$ is the average integrated intensity of ferrite peaks. The lattice parameters of RA were evaluated according to the Nelson-Riley method from the highest γ diffraction peak (200) [18]. The carbon content in as-quenched and partitioned austenite can be calculated by [19]
(2)$$a\;=\;0.3572\;+\;0.00045\;C\;+\;0.00009{\;\mathrm{Cr}}_{\gamma }$$
where *a* is the austenite lattice parameter, nm, C is carbon content and Cr_γ_ is chromium content in at.% in austenite. The calculated results are converted to mass percentage (wt.%) for further discussion.

### 2.2. Microstructural Characterization

The alloying element distribution in samples after polishing was observed using field-emission electron probe micro analyzer (EPMA, JEOL JXA-8530F, Tokyo, Japan), operating at 20 kV accelerating voltage. Metallographic samples were etched for 10 s with a solution of 20 g FeCl_3_, 25 mL hydrochloric acid and 200 mL distilled water. The microstructure and impact fracture surface were studied by field-emission scanning electron microscope (FE-SEM, ULTRA 55, Zeiss, Jena, Germany) operating at 20 kV. The semi-quantification of chemical composition in the precipitates was performed by energy disperse spectroscopy (EDS) as equipped in SEM. The backscattered electron (BSE) images are observed directly from the mechanically polished samples but without etched samples, which can prevent the carbides shedding or being lost during etching. The area fraction of the undissolved carbides is measured using the Image J software (Version 1.53e, National Institute of Health, Bethesda, MD, USA). The average value of the area fraction of carbides was calculated from ten BSE images. Electron backscatter diffraction (EBSD) (Symmetry, Oxford Instruments, Oxford, UK) maps were obtained with a scanning step size of 50 nm. Data acquisition and post-processing were performed using the AZtec software (version 4.3, Oxford Instruments, Oxford, UK). EBSD sam ples were electro-polished for 20 s with 8% perchloric acid and 92% ethanol solution at 20 °C with a voltage of 18–22 V, after mechanical polishing. Transmission electron microscopy (TEM) was conducted using a field-emission TEM (FE-TEM, Tecnai G^2^ F20, FEI, Hillsboro, OR, USA), operated at 200 kV. Thin-foil TEM samples were prepared by twin-jet-polishing with the same solution of electro-polishing at −20 °C after being ground to 40 μm and punched to a 3-diameter disc.

### 2.3. Mechanical Properties Measurement

Tensile test samples were cut by electrical-discharge machine (EDM) according to ASTM standard [20] (gauge length of 50 mm, gauge width of 12.5 mm and gauge thickness of 2 mm). They were obtained from the 1/4 thickness location of the 5.5 mm-thick sheet. Tensile tests were conducted using the Instron tensile testing machine at a constant crosshead speed of 2 mm min^−1^. The average value of the tensile test results was calculated from three samples for each case. Interrupted tensile tests were conducted to measure the volume fraction of transformed austenite induced by strain. The Charpy impact energy was measured on standard Charpy U-notch specimens (5 mm × 10 mm × 55 mm, 2 mm in notch depth) using the ZBC2452-B pendulum impact testing machine (MTS, Shanghai, China) at room temperature. The notch was parallel to the rolling direction, and the average value was obtained based on three samples for each condition.

## 3. Results

### 3.1. Phase Diagram 

The equilibrium phase diagram and carbon and chromium content in austenite with the function of temperature, calculated by the Thermo-Calc software (version 2017b, Stockholm, Sweden) with the TCFE9 database [21], are shown in Figure 1. M_23_C_6_ (M = Cr, Fe) and MC (M = Nb, Ti) carbides still exist at *A*_e3_ (798 °C), i.e., 0.0346 mole and 0.0018 mole, respectively, as can be seen in Figure 1a. With the temperature increases to 974 °C, the M_23_C_6_ can be completely dissolved while the dissolution of MC is negligible. Thus, the carbon and chromium content in austenite increase gradually due to the dissolution of M_23_C_6_, as can be seen in Figure 1b. The black dash line in Figure 1b represents the bulk carbon content in the studied alloy. As the temperature is above 974 °C, the carbon in austenite is approximately 0.22 wt.%, at which no M_23_C_6_ exists. From this, the depleted carbon content caused by MC is determined to be approximately 0.01 wt.%. 

The measured *A*_c1_ and *A*_c3_ are 836 and 953 °C, respectively, as can be seen in Figure 2a. This difference between the measured *A*_c3_ and calculated *A*_e3_ may be caused by the heating rate [22]. With the austenitizing temperature increase from 1000 to 1060 °C, *M*_s_ decreases from 194 to 170 °C, as can be seen in Figure 2b. The carbon diffusion distance in austenite caused by auto-tempering during quenching is evaluated to be 2.73 × 10^−1^ nm, considering the period from such low *M*_s_ to room temperature with a cooling rate of 30 °C/s [23,24], which is less than the lattice parameters of FCC iron [25], and thus it can be ignored, which indicates that the carbon content of as-quenched martensite and austenite is identical to that of prior austenite before quenching, where the prior austenite and as-quenched austenite represent the formed austenite phase during austenitizing and untransformed austenite during quenching, respectively.

### 3.2. As-Quenched and Partitioned Microstructure

From the BSE and corresponding alloying element distribution maps, the carbides, indicated by white arrows in the BSE images, are spherical or elliptic in samples 1000-Q~1060-Q, as can be seen in Figure 3. With the increasing austenitizing temperature, large Cr-rich carbides (M_23_C_6_) disappear gradually, and no Cr-rich carbides were observed in the 1060-Q sample. However, small Ti or Nb-rich carbides (MC) are quite stable in all samples, as can be seen in Figure 3. These findings are consistent with the equilibrium calculation, as can be seen in Figure 1a. Therefore, according to the alloying elements distribution maps, together with *M*_s_ calculation Equation [11]: $M_{s}\;=\;539-423C-30.4\mathrm{Mn}-12.1\mathrm{Cr}-17.7\mathrm{Ni}-7.5\mathrm{Mo}+10\mathrm{Co}-7.5\mathrm{Si}$ (wt.%), the reduction of solute chromium and carbon in prior austenite is the main reason for the increase in *M*_s_ in the 1000-Q sample, i.e., 24 °C, as can be seen in Figure 2b.

The microstructures of 1000-P–1060-P are mainly lath martensite with distributed secondary carbides, as can be seen in Figure 4a–d. As can be seen in Figure 4, spherical or elliptic submicron-sized carbides are mainly located at the prior austenite grain boundary (PAGB) and a few are in the matrix—nano-sized spherical carbides are found in all samples. These carbides are identified as M_23_C_6_ and MC by combination of SAED and EDS (due to alloying elements difference), as can be seen in Appendix A in the Appendix A. As the austenitizing temperature increases, the content of M_23_C_6_ decreases, and it is rare to observe M_23_C_6_ at 1060 °C, which is consistent with Figure 1a. Considering the very low diffusivity of substitutional atoms at 250 °C, the aforementioned two types of carbides will not growth or shrink significantly during partitioning. It should be noted that the RA is very difficult to distinguish from other features due to its very fine size and weak contrast in SEM, and thus a detailed analysis will be given late by EBSD.

The depicted PAGBs of 1000-P~1060-P are shown in Figure 5. To analyse the prior austenite grain sizes (PAGSs), a mean linear intercept method is applied. With the austenitizing temperature increasing from 1000 to 1060 °C, the PAGS of the samples 1000-P, 1020-P, 1040-P, 1060-P are 11.4 ± 1.3 μm, 12.9 ± 0.6 μm, 15.4 ± 0.7 μm and 18.4 ± 3.8 μm, respectively. The PAGS increases with increasing austenitizing temperature, which is mainly related to its high activation energy and weak pinning effect of secondary carbides at elevated temperature [26]. Furthermore, the grain growth in the present alloys is significantly restrained due to the undissolved M_23_C_6_ and MC, as compared with the PAGS in references [6,26].

With increasing austenitizing temperature, the volume fraction of RA in as-quenched and partitioned samples increases gradually, as can be seen in Figure 6a, which is the result of high carbon and chromium content in the matrix caused by a gradual dissolution of M_23_C_6_. The RA in 1060-NP is much lower than that in 1060-P, which is related to the stability of austenite varying with temperature [27]. Relative to as-quenched samples, the increased RA content in 1040-P and 1060-P may be attributed to carbon atoms segregating at the martensite-austenite boundary (austenite reversion) during partitioning [28]. The equilibrium Cr: C ratio (at.%) in M_23_C_6_ is approximately 49: 20 at the temperature range of 1000~1040 °C [21], as can be seen in Appendix A in the Appendix A, and the depleted carbon by MC is approximately 0.05 at.% (0.01 wt.%) at the present temperature range, as can be seen in Figure 1. As a result, the depleted chromium and carbon in the alloy should obey the following equation:(3)$${\mathrm{Cr}}_{A}{-\;\mathrm{Cr}}_{\gamma }{\;=\;49(C}_{A}\;-\;0.05{\;-\;C}_{\gamma })/20$$
where Cr_A_ is the chromium content (13 at.%) in the bulk alloy, Cr_γ_ is chromium content in prior austenite, C_A_ is carbon content (1.04 at.%) in the bulk alloy, C_γ_ is the carbon content in prior austenite, ‘${\mathrm{Cr}}_{A}{-\;\mathrm{Cr}}_{\gamma }$’ and ‘$C_{A}\;-\;0.05{\;-\;C}_{\gamma }$’ represent the chromium and carbon contents (at.%) depleted by M_23_C_6_, respectively. Assuming no significant auto-tempering occurs, the chemical composition of the as-quenched martensite and austenite will inherit from prior austenite. Therefore, the chromium and carbon contents in as-quenched martensite as well as austenite can be calculated after combining Equations (2) and (3). The carbon content in as-quenched martensite as well as austenite increases with increasing austenitizing temperature, as can be seen in Figure 6b. During partitioning at 250 °C, the movement of substitutional atoms (i.e., chromium, manganese) can be neglected, according to the “constrained carbon equilibrium” (CCE) model [29,30]. The calculated carbon content in the RA of partitioned samples is 0.66 ± 0.08, 0.79 ± 0.06, 0.92 ± 0.06 and 0.85 ± 0.1 (wt.%) in 1000-P, 1020-P, 1040-P and 1060-P, respectively. As previously noted, the RA in 1060-NP is close to that in 1000-P, by which the influence of carbides could be determined. Meanwhile, the higher RA carbon content in 1060-NP may be attributed to the lower volume fraction of RA and austenite reversion at carbon-rich boundaries [28].

In addition to the typical low temperature martensitic microstructure, blocky submicron-sized RA is also observed at various boundaries, but no clear conclusion can be drawn on the distribution of RA in all the partitioned samples, see Figure 7a–d. The amount as well as the size of blocky RA increases with increasing austenitizing temperature, as can be seen in Figure 7e. Typical lath martensite with high density dislocation is mainly microstructural in all the partitioned samples, as can be seen in Figure 8, which are consistent with the SEM and EBSD observations. Furthermore, intra lath RA, tens of nanometers in width, which cannot be detected in SEM-EBSD due to the limitation of the beam interaction volume in SEM, was also found in all the partitioned samples, as can be seen in Figure 8. Noticeably, no obvious large ε-carbides and cementite were found in any of the samples even though careful searching was applied to the whole region under TEM field of view.

To quantitatively analyze the microstructural variations compared to the austenitizing temperature, the volume fraction and carbon content of RA, the area fraction of M_23_C_6_ as well as the chromium and carbon content in the matrix are listed in Table 1, where the microstructure variations in 1060-NP and 1060-P caused by the quenching temperature are listed as well. The area fraction of M_23_C_6_ in 1000-P~1040-P can be estimated by subtracting the area of MC in 1060-P from the total area fraction of carbides. It is assumed that except for part of the carbon partitioning to austenite, the retained carbon atoms are solid solution in martensite [29]. Therefore, the average carbon content in tempered martensite can be estimated by
(4)$$C_{T\alpha }{\;=\;(C}_{\gamma }{\;-\;C}_{P\gamma }{\;\times \;V}_{P\gamma \;}{/100)/(1-V}_{P\gamma \;}/100)$$
where C_Tα_ and C_Pγ_ represent the carbon content in tempered martensite and partitioned austenite, respectively, where the partitioned austenite represents the untransformed austenite during partitioning and secondary cooling. V_Pγ_ represents the volume fraction of RA in the partitioned samples. 

### 3.3. Mechanical Properties 

The engineering stress-strain curves and impact toughness of the partitioned samples are shown in Figure 9. The 0.2% offset strain load is defined as yield strength (YS). The YS varies from 1237 ± 19 to 1290 ± 18 MPa as the austenitizing temperature increases from 1000 to 1060 °C, and 1000-P has the highest YS, as can be seen in Table 2. In view of M_23_C_6_ carbides being submicron-sized in the three samples, the contribution of precipitation strengthening on YS is very weak, less than 5 MPa, and could thus be ignored [31,32]. Hence, the highest YS of 1000-P is mainly due to the finest grain and lowest fraction of the soft phase RA among all the partitioned samples [33,34,35]. With increasing austenitizing temperature, the ultimate tensile strength (UTS) of samples 1000-P~1060-P increases gradually, while the uniform elongation (UE) and total elongation (TE) increase first and then decrease (Table 2). However, the impact toughness increases from 53.3 ± 10.2 to 94.6 ± 8.0 Jcm^−2^ continuously with increasing austenitizing temperature, as shown in Figure 9b and Table 2. To clarify the difference caused by the austenitizing temperature, the deceiving factors on ductility and impact toughness will be further discussed in the late section.

## 4. Discussion.

### 4.1. Microstructure Evolution during Quenching and Partitioning

The schematic of the microstructure evolution during quenching and partitioning is shown in Figure 10, aiming to elucidate the microstructure evolution during the whole process of MCMSSs. Clearly, with the austenitizing temperature increases, the microstructure changes significantly, see Figure 10a–f. At low austenitizing temperature (e.g., 1000 °C), M_23_C_6_ as well as MC cannot be completely dissolved, by which part of the carbon and chromium is depleted in prior austenite, as shown in Figure 10a. Then, the M_23_C_6_ and MC will be inherited from prior austenite to the as-quenched microstructure, i.e., fresh martensite and RA, as shown in Figure 10b. During partitioning, carbon atoms diffuse from supersaturated martensite to RA, leading to a carbon increase in RA but decreases in martensite, as shown in Figure 10c as well as Figure 6b. As the austenitizing temperature increases, the M_23_C_6_ is gradually dissolved and then completely disappears, leading to an increasing carbon and chromium content in the prior austenite, as shown in Figure 10d. Meanwhile, PAGS increases slightly due to the reduced pinning effects of secondary carbides. Compared with the microstructure at lower austenitizing temperature, the volume fraction of RA increases after quenching, which is due to the carbon and chromium content remains in prior austenite, as shown in Figure 10b,e, and the available carbon content for partitioning increases, as shown in Figure 10b,d–f.

When it comes to the carbon content in RA after partitioning, the average carbon diffusion distance *x* in austenite is estimated using the simplified Equations (5) and (6) [23,24]:(5)$$x^{2}\;=\;6Dt\;$$
(6)$${D\;=\;D}_{0}\exp(-\frac{Q}{\mathrm{RT}})$$
where *D* is the carbon diffusion coefficient in FCC iron, *t* is the diffusion time in seconds, D_0_ is a constant factor (here, D_0_ = 1.5 × 10^−5^ m^2^ s^−1^) [24], Q is the activation energy for carbon diffusion (Q_γ_ = 1.421 × 10^5^ J mol^−1^) [24], T is the absolute temperature in Kelvin and R is the gas constant (8.314J mol^−1^ K^−1^). The calculated carbon diffusion distance after partitioning at 250 °C (523 K) for 30 min is 32.2 nm, which is smaller than the half width (or radius) of submicron-size blocky RA. This indicates that the distribution of carbon in the submicron-sized blocky RA is not homogeneous, i.e., the carbon partitioning is insufficient. It is also noteworthy that the size of RA in 1040-P is smaller than that in 1060-P, as shown in Figure 7. The carbon atoms diffusion is faster in the fine RA grains than that in the coarse ones [25,36], which may be the reason for a slightly higher carbon content of RA in 1040-P than in 1060-P. However, the carbon content of RA in 1000-P and 1020-P are relatively lower than that in 1040-P, which is mainly due to the decreasing amount of available carbon for partitioning in as-quenched martensite. 

### 4.2. Effects of Microstructure on Ductility and Impact Toughness

In general, the effective grain size (i.e., packet, block or sub-block width) in martensite decreases with the decreasing PAGS [37,38]. The grain refinement within a certain range is always followed by both improving the toughness and ductility of steels [39,40]. However, this is conflicted with the present findings, as shown in Figure 5 and Table 2. Thus, the microstructure features (chemical composition in the matrix, RA, morphology and size of carbides) other than grain size play a dominant role in improving the toughness and ductility of the present alloys. 

#### 4.2.1. Effect of Microstructure on Ductility

The ductility is represented by UE as well as TE during the tensile test in the present case. When comes close to 1000-P and 1060-NP, the UE of 1000-P and 1060-NP is close to each other while their TE is different, that is, the TE of 1000-P is much higher than that of 1060-NP, as shown in Table 2 and Figure 11a. It is also worth noting that the ductility difference caused by the RA in 1000-P and 1060-NP is very small due to its very low fraction, as shown in Table 2, even though their mechanical stability may be different due to their chemical compositions [41]. 

Considering their large size, the contribution of M_23_C_6_ carbides to work hardening caused by inhibiting the dislocation motion can be ignored [31,32]. Thus, the role of M_23_C_6_ carbides in improving UE is insignificant as well. At the uniform deformation stage, the deformation occurs preferentially at the area where lower flow stress is needed and the local stress concentration at M_23_C_6_ carbides increases slowly. As a result, it is difficult for a void nucleate nearby M_23_C_6_ carbides [42], which is consistent with the obtained UE in 1000-P and 1060-NP. Thereafter, at the necking instability stage, the local stress concentration increases quickly at the M_23_C_6_/matrix interface, and naturally, TE will be deteriorated by the appearance of M_23_C_6_ carbides in 1000-P if no other feature is involved. However, the present finding is as shown in Table 2 and Figure 11a is the opposite with the aforementioned discussion.

It is reported that UE tends to increase with increasing carbon content, which has been explained by high carbon content, which leads to a high degree of work hardening rate [43]. The carbon/chromium content (C_Tα_ = 0.18, Cr_γ_ = 12.5 wt.%) of the matrix in 1060-NP is much higher than that (C_Tα_ = 0.11, Cr_γ_ = 11.6 wt.%) in 1000-P, as shown in Table 2. Furthermore, the work hardening rate of 1060-NP is slightly higher than that of 1000-P at the initial stage (below 0.02), and then is similar to that of 1000-P, as shown in Figure 11b. This indicates that the effects of carbon/chromium content improving UE, especially at the later stage, are also insignificant in this study. The higher the work hardening rate is, the higher the flow stress is [43], and then critical triaxial stresses for void nucleation can be achieved by a small necking deformation [43,44]. Furthermore, when the necking instability occurs, the flow stress of 1060-NP is much higher than that of 1000-P, as shown in Figure 11c, resulting in a small deformation until fracture occurs, which has been observed in 4330, 4340, and 4350 steels as well [43]. Thus, the decreased matrix carbon/chromium content can improve ductility by increasing the necking strain needed for void nucleation. 

In addition, the RA plays an important role in improving UE and TE in stainless steels due to transformation-induced plasticity (TRIP) effects [10,11,28]. The variations between 1060-NP and 1060-P are shown in Table 1. According to the previous discussion, the difference in UE caused by the matrix carbon/chromium content can be ignored. The UTS of 1060-NP is higher than that of 1060-P, but the UE and TE are much lower, see Figure 12a and Table 2. When the volume fraction of RA increases from 3.2 ± 0.4% to 10.8 ± 0.5%, the UE and TE of 1060-P increase by 1.0 ± 0.1% and 3.2 ± 0.3%, respectively. In detail, (a) during the uniform deformation stage, the transformed RA in 1060-NP and 1060-P is 1.5 ± 0.2% and 7.5 ± 0.6%, respectively, see Figure 12b; (b) during necking deformation, the transformed austenite is 0.5 ± 0.2% and 2.3 ± 0.3%, respectively. The volume fraction of RA decreases with increasing in strain ε, as described [17]:(7)$${\ln V}_{\gamma }{-\ln V}_{\gamma_{0}}=\;-k\varepsilon$$
where $V_{\gamma_{0}}$ is the volume fraction of RA before deformation, *k* is a constant. A lower *k* value represents higher mechanical stability of RA. The calculated *k* value of 1060-NP and 1060-P is 14.5 and 24.3, respectively, which are in the ranges of reported values [41]. This indicates that the mechanical stability of RA in 1060-NP is much higher than that in 1060-P. It should be also noted that the volume fraction of RA in 1060-NP is too low to provide a sound TRIP effect during deformation, and then result in a lower ductility than that of 1060-P.

Furthermore, the ductility of 1040-P is higher than that of 1060-P, even though a relative low fraction of RA was obtained, as can be seen in Table 2. When the engineering strain is 7.9%, the transformed austenite content is 7.1 ± 0.4% in 1040-P, which is slightly lower than that in 1060-P, see Figure 12b. As can be seen, the *k* value of 1040-P is 20, which indicates that the mechanical stability of RA in 1040-P is higher than that of 1060-P. The high mechanical stability of RA in 1040-P is mainly attributed to its higher carbon content and finer grain size, as can be seen in Figure 6b and Figure 7e. These results indicate that the higher mechanical stability of RA is another dominant reason for improving ductility.

#### 4.2.2. Effect of Microstructure on Impact Toughness

According to the relationship between the volume fraction of RA and impact toughness of all partitioned samples, as given in Figure 13, the increasing impact toughness does not follow a linear relationship with the volume fraction of RA. Compared with that of 1060-NP, the impact toughness of 1020-P and 1040-P decreases when the volume fraction of RA increases. This indicates that the impact toughness does not only depend on the volume fraction of RA. Concerning on the mechanical stability of RA, the volume fraction of RA in 1040-P is comparable with that of 1060-P, but the mechanical stability of RA is higher. Accordingly, the impact toughness of 1040-P and 1060-P is expected to be comparable, but this is not the case, as can be seen in Figure 13. This suggested that RA does not play a decisive role in impact toughness. However, the RA on improving impact toughness has been extensively studied in low alloy steels [45,46,47], which attribute its improvement to its crack-arrest and stress-relief effect. Gao [46] suggested that low stable blocky RA could promote the formation of a second crack due to the increased incompatibility between hard fresh martensite and the surrounding matrix during deformation, but the film-like RA with higher stability can hinder the crack propagation more efficiently. The TRIP effects of RA on impact toughness include 1) the energy dissipation via strain-induced martensite transformation and 2) the energy reduction for crack propagation caused by fresh martensite [48]. The TRIP effects can be beneficial on improving impact toughness only when the energy dissipation is higher than the energy reduction. In the present case, the energy dissipation by submicron-sized blocky RA may not overcome the energy reduction during the impact test.

For the impact toughness test, the fracture model depends on the flow stress of martensite. When the flow stress is low, a large amount of plastic deformation is required before fracture, i.e., ductile fracture. When the flow stress is higher than the fracture stress, the brittle fracture may occur [49]. The carbon depletion in martensite can reduce flow stress by two mechanisms. One is that the lattice distortion caused by supersaturated carbon atoms decreases [50], and the other is the interaction of dislocation and carbon atoms by impeding screw dislocations from cross slipping and multiplying becomes weak [49,51]. In general, the increase in carbon/chromium content in matrix reduces the impact toughness by increasing flow stress. Surprisingly, the results are opposite, as can be seen in Figure 14a, which indicates that the impact toughness should be determined by the other factors. 

The fine precipitates, e.g., (Nb, Ti, V)C carbides, can improve toughness by pinning austenite grain boundaries to limit grain growth [52], but the large precipitates, e.g., TiN (>0.5 μm), will promote crack initiation and propagation [53]. The submicron-size M_23_C_6_ carbides are mainly located at the grain boundary, as can be seen in Figure 5, which can easily reduce the interface cohesion and cause crack nucleation. The impact fracture morphologies of 1000-P, 1020-P and 1040-P, as can be seen in Figure 15a–c, and the corresponding dispersed energy of undissolved M_23_C_6_ carbides, as can be seen in Figure 15d–f, indicate that the coarse M_23_C_6_ carbides are the sites for crack nucleation. The impact toughness increases markedly with the decreasing area fraction of M_23_C_6_, as can be seen in Figure 14b, which further indicates that the undissolved M_23_C_6_ carbides play a decisive role on impact toughness.

### 4.3. The Different Microstructure Effect on Tensile and Impact Tests

The sensitivity of impact toughness in carbides is much higher than that of tensile properties [54,55]. In the present work, the M_23_C_6_ carbides have a big deterioration effect on the impact toughness, but a weak effect on tensile properties. At the same time, the metastable RA has a strong effect on the ductility, but the roles of RA (volume fraction and stability) on toughness are slight. The differences in deformation mechanism between the impact and tensile tests could explain the above experimental findings.

The main difference between the impact toughness test and the tensile test contains two sides. One is the deformation velocity of the different tests. The velocity of a quasi-static tensile test is 3.3 × 10^−5^ m s^−1^, while that of an impact test is 5.2 m s^−1^ in this study. It is reported that the flow-stress can be significantly increased by increasing the strain rate [56], and a higher strain-rate leads to a higher resistance to glide of screw dislocation [57]. The other one is the dimension of samples. The deformation area of tensile samples is uniform and smooth, but that of impact samples is notched. The deformation of tensile includes a uniform and necking stage, and the sharply increased stress concentration only occurs at the necking stage. However, the impact deformation is local, and the stress concentration ahead of the tip always increases quickly. Thus, during the impact test, the quick increased stress concentration at the heterogeneous phase (i.e., fresh martensite and undissolved carbides), caused by high flow stress, may cause the crack nucleation even at a very small strain stage. The TRIP effect is not so effective at such a type test. Inversely, during the tensile test, the flow stress increases slowly, and the stress concentration can be relieved effectively by TRIP effects or uniform deformation [58].

## 5. Conclusions

The effects of austenitizing temperature tempering on the microstructure of MCMSSs have been systematically studied and discussed in terms of tensile and impact properties, and the following conclusions are drawn: Typical martensitic microstructure together with blocky RA and distributed carbides are observed in all partitioned alloys. The undissolved M_23_C_6_ carbides are spherical or elliptic with a submicron size, and mostly located at the prior austenite grain boundary. The prior austenite grain size (PAGS) increases slowly with increasing austenitizing temperature, and the grain growth can be restrained by the pinning effects of fine MC carbides. In addition, the elements (C, Cr) in the matrix content and volume fraction of RA increase with the gradual dissolution of M_23_C_6_.With the increasing austenitizing temperature from 1000 to 1060 °C, the ductility increases first and then decreases, but the impact toughness keeps increasing significantly. By optimizing austenitizing temperature (1060 °C for 40 min) and tempering (250 °C for 30 min) heat treatments, the MCMSSs demonstrate excellent mechanical properties with the ultimate tensile strength of 1740 ± 8 MPa, a yield strength of 1237 ± 19 MPa, total elongation of 10.3 ± 0.7 % and impact toughness of 94.6 ± 8.0 Jcm^−2^ at room temperature.The improving ductility is attributed to a synergic cooperation between the volume fraction and mechanical stability of RA. The depleted carbon and chromium content in the matrix also contribute to ductility by increasing the necking deformation needed for void nucleation. However, the undissolved M_23_C_6_ carbides decrease the impact toughness dramatically by acting as a crack initiator.

## Figures and Tables

**Figure 1 materials-14-01000-f001:**
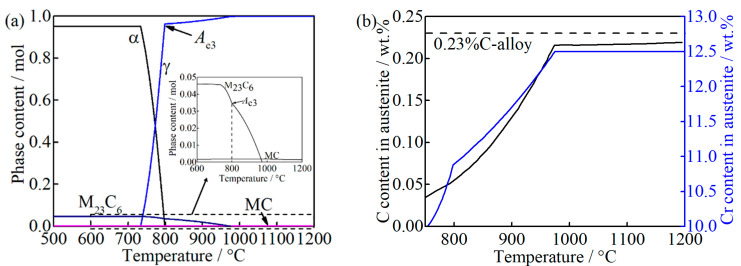
Thermodynamic calculations of (**a**) the equilibrium phases diagram; (**b**) the carbon and chromium content in austenite as a function of temperature, where “M_23_C_6_” represents (Cr, Fe)_23_C_6_, and “MC” represents (Nb, Ti) C.

**Figure 2 materials-14-01000-f002:**
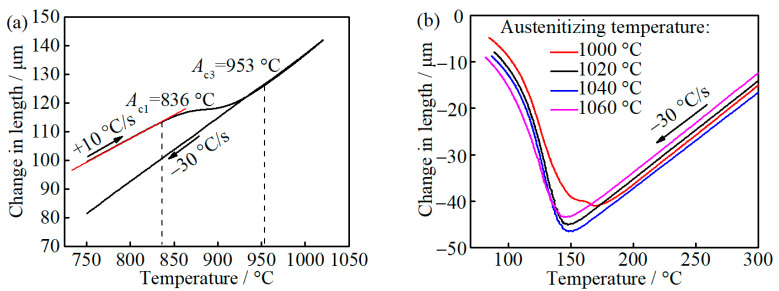
The relative length change of dilatometric samples with temperature: (**a**) heating and cooling; and (**b**) cooling from different austenitizing temperature.

**Figure 3 materials-14-01000-f003:**
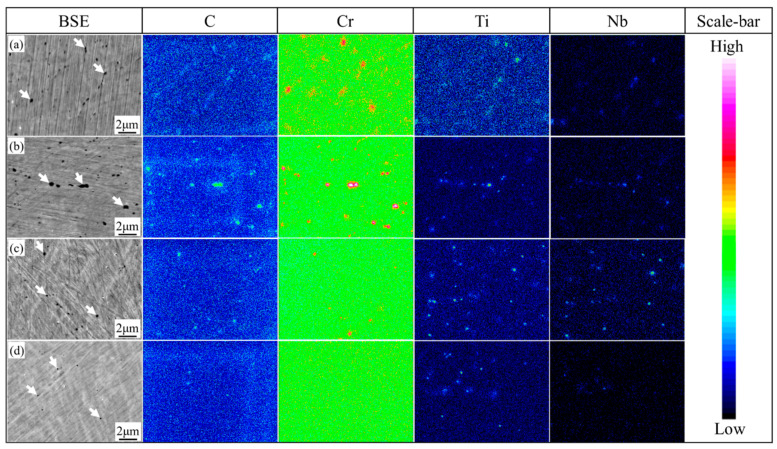
Alloying elements distribution maps of (**a**) 1000-Q; (**b**) 1020-Q; (**c**) 1040-Q; and (**d**) 1060-Q, where white arrows represent undissolved carbides and the scale bar indicates the concentration of each element.

**Figure 4 materials-14-01000-f004:**
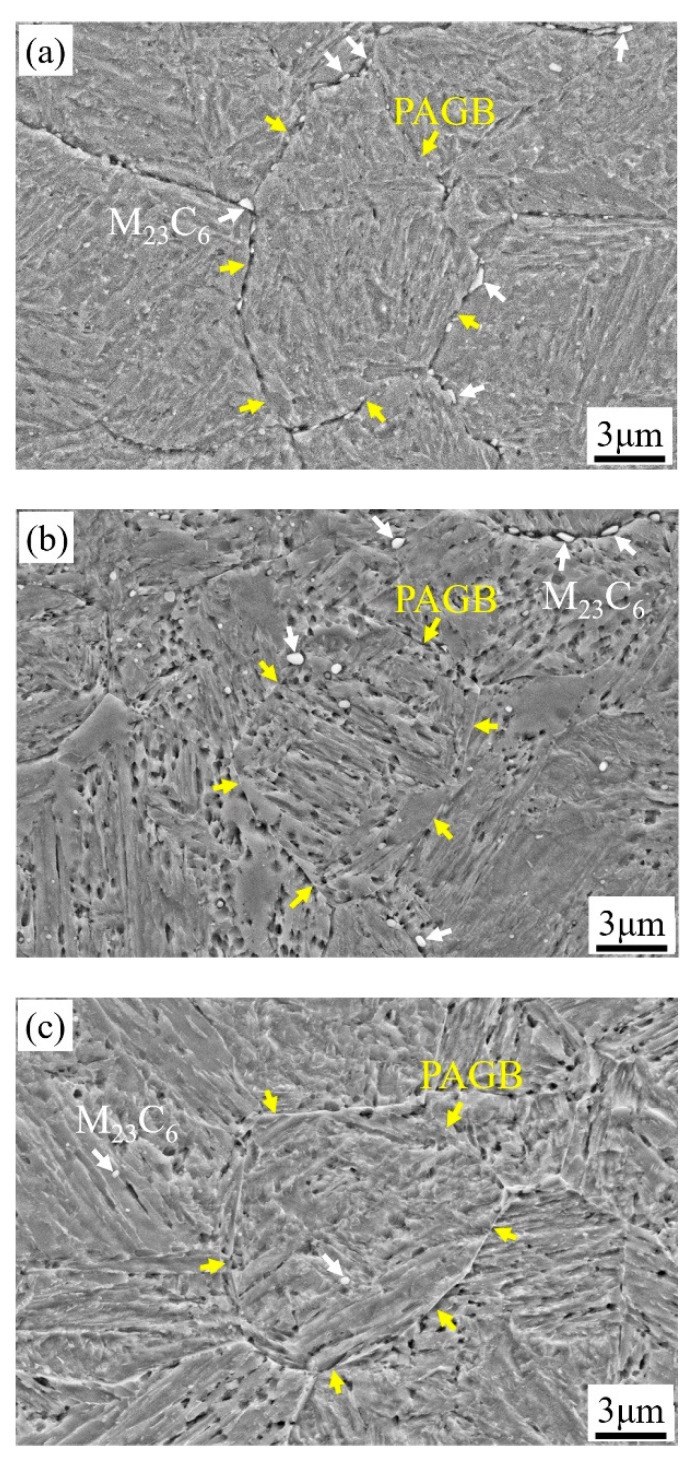
SEM images of (**a**) 1000-P; (**b**) 1020-P; (**c**) 1040-P; (**d**) 1060-P (white, red and yellow arrows represent the M_23_C_6_ carbides, MC carbides and the prior austenite grain boundary (PAGB), respectively).

**Figure 5 materials-14-01000-f005:**
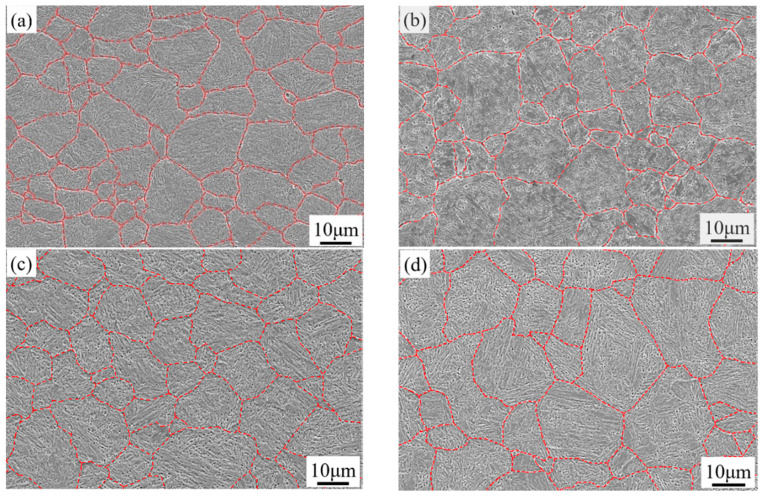
The prior austenite grain boundary of (**a**) 1000-P; (**b**) 1020-P; (**c**) 1040-P; and (**d**) 1060-P.

**Figure 6 materials-14-01000-f006:**
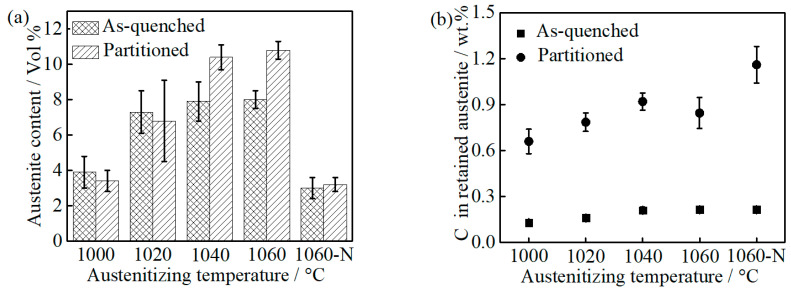
(**a**) Retained austenite content and (**b**) the carbon content in the retained austenite of as-quenched and partitioned samples.

**Figure 7 materials-14-01000-f007:**
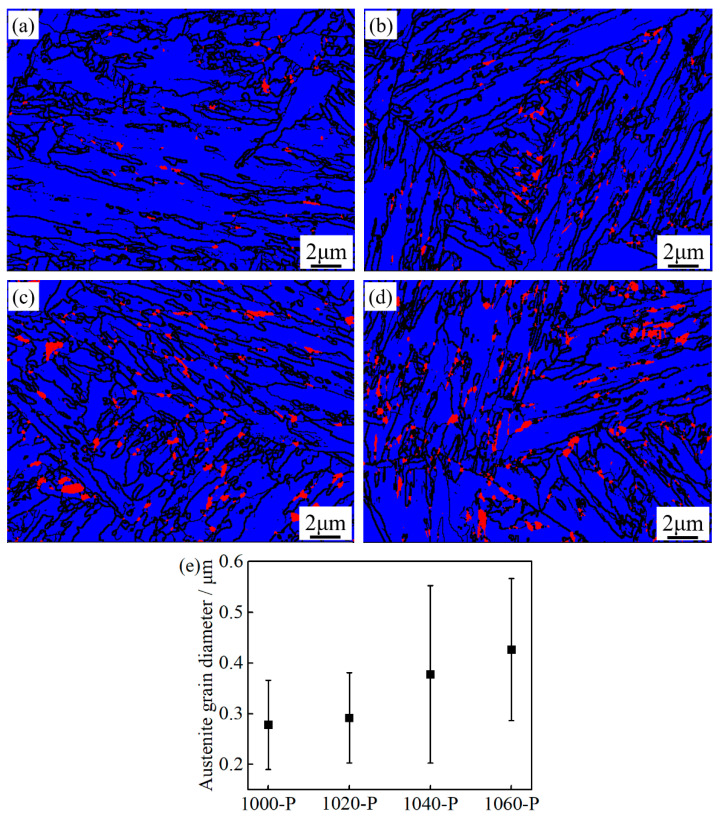
Combined phase map and grain boundaries of (**a**) 1000-P; (**b**) 1020-P; (**c**) 1040-P; and (**d**) 1060-P (red, blue and black represent the retained austenite, martensite and boundaries with misorientation >10°, respectively); (**e**) average equivalent circle diameter of the retained austenite (RA).

**Figure 8 materials-14-01000-f008:**
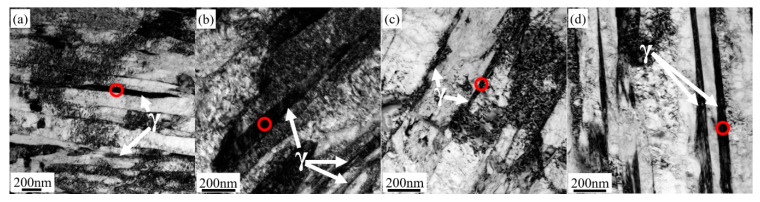
TEM bright field images of retained austenite in (**a**) 1000-P; (**b**) 1020-P; (**c**) 1040-P; (**d**) 1060-P, corresponding dark Figure (**e**) 1000-P; (**f**) 1020-P; (**g**) 1040-P; and (**h**) 1060-P, were the red annulus area represents the selected area diffraction pattern (SAED).

**Figure 9 materials-14-01000-f009:**
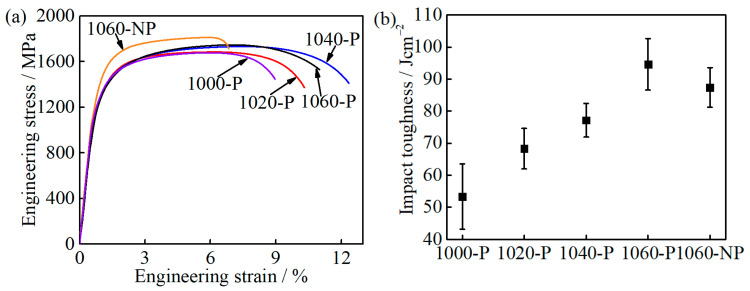
Mechanical properties of the partitioned samples: (**a**) engineering stress-strain curves; and (**b**) impact toughness.

**Figure 10 materials-14-01000-f010:**
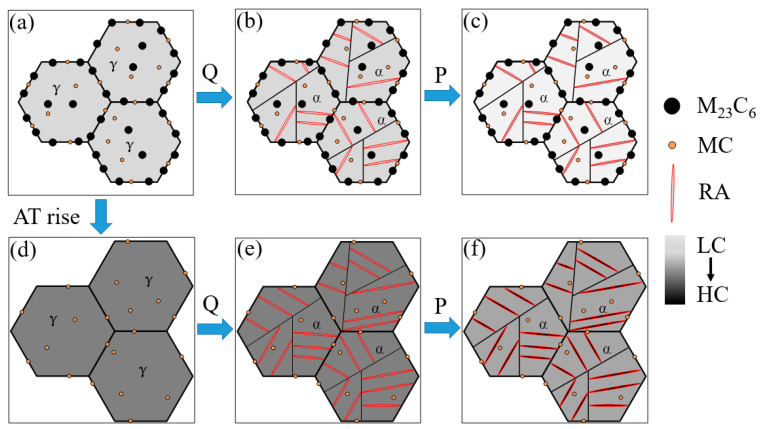
The schematic of the microstructure evolution during the quenching and partitioning at different austenitizing temperatures, (**a**,**d**) austenitizing, (**b**,**e**) quenching, (**c**,**f**) partitioning, where “AT” represents the austenitizing temperature, “LC” and “HC” represent the low and high carbon/chromium content, respectively, and the darker the color is, the higher the content (the chromium content in quenched and partitioned martensite/austenite is identical).

**Figure 11 materials-14-01000-f011:**
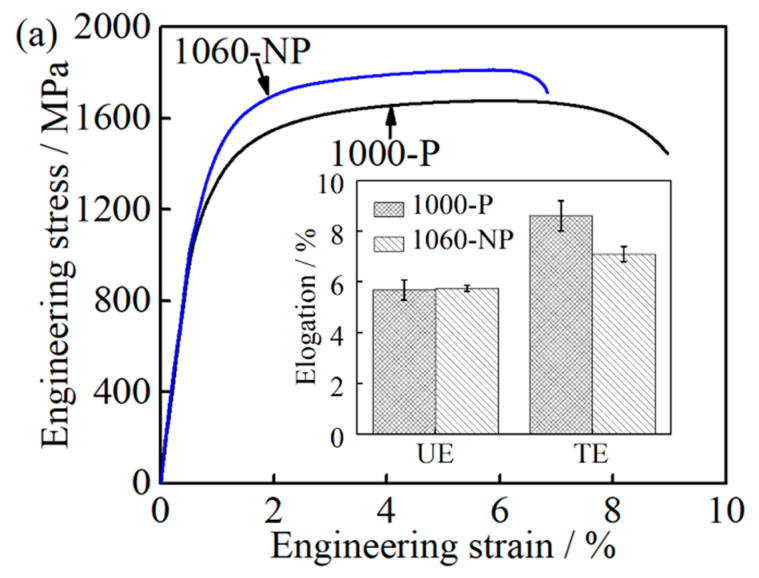
Comparison between the 1000-P and 1060-NP: (**a**) engineering stress-strain curves and elongation; (**b**) work hardening curves; and (**c**) the true stress-strain curves.

**Figure 12 materials-14-01000-f012:**
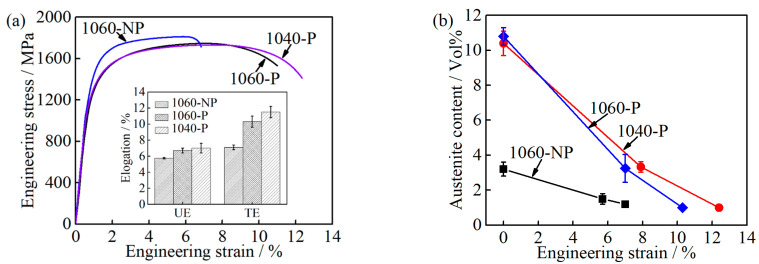
Comparison among 1060-NP, 1060-P and 1040-P (**a**) engineering stress-strain curves and elongation; and (**b**) austenite content at a different engineering strain.

**Figure 13 materials-14-01000-f013:**
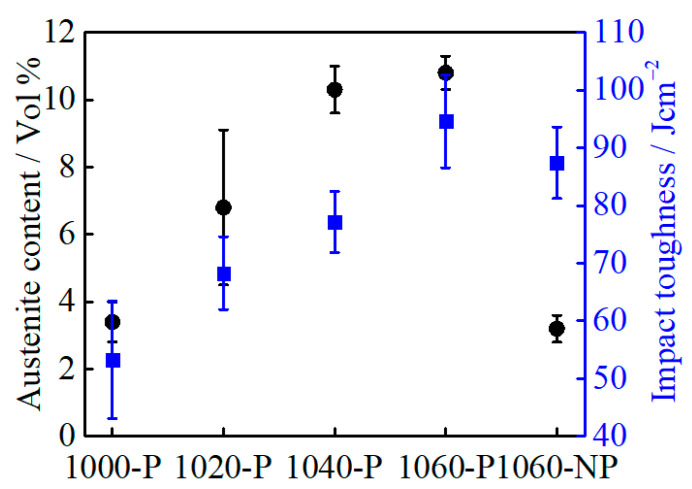
The relationship between retained austenite volume fraction and impact toughness.

**Figure 14 materials-14-01000-f014:**
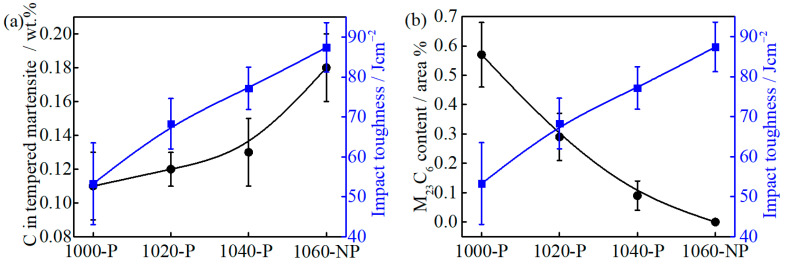
The relationship between (**a**) tempered martensite carbon content and impact toughness; and (**b**) M_23_C_6_ area fraction and impact toughness.

**Figure 15 materials-14-01000-f015:**
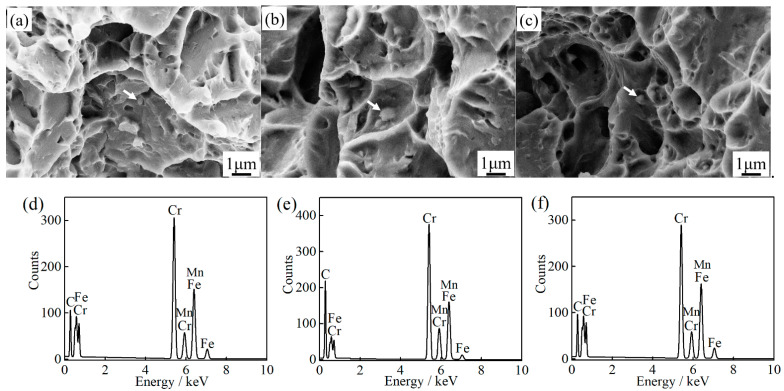
The impact fracture morphology of (**a**) 1000-P; (**b**) 1020-P; (**c**) 1040-P, with the corresponding dispersed energy of carbides (the area showed by white arrow); (**d**) 1000-P; (**e**) 1020-P; and (**f**) 1040-P.

**Table 1 materials-14-01000-t001:** Volume fraction of RA, elements (C, Cr) content and the area fraction of M_23_C_6._

Samples	V_Pγ_ (%)	C_Pγ_ (wt.%)	C_Tα_ (wt.%)	Cr_γ_ (wt.%)	A_M23C6_ (%)
1000-P	3.4 ± 0.6	0.66 ± 0.08	0.11 ± 0.02	11.6 ± 0.2	0.57 ± 0.11
1020-P	6.8 ± 2.3	0.79 ± 0.06	0.12 ± 0.01	11.9 ± 0.3	0.29 ± 0.08
1040-P	10.3 ± 0.7	0.92 ± 0.06	0.13 ± 0.02	12.4 ± 0.1	0.09 ± 0.05
1060-P	10.8 ± 0.5	0.85 ± 0.1	0.14 ± 0.01	12.5	0
1060-NP	3.2 ± 0.4	1.16 ± 0.12	0.18 ± 0.02	12.5	0

The terms of V_Pγ_, C_Pγ_, C_Tα_, Cr_γ_ and A_M23C6_ stand for the RA volume fraction in the partitioned samples, carbon content in the partitioned austenite, the carbon content in tempered martensite, the chromium content in prior austenite and the area fraction of M_23_C_6_ carbides, respectively.

**Table 2 materials-14-01000-t002:** The results of the tensile and impact tests.

Samples	YS (MPa)	UTS (MPa)	UE (%)	TE (%)	IT (Jcm^−2^)
1000-P	1292 ± 18	1677 ± 2	5.7 ± 0.4	8.6 ± 0.6	53.3 ± 10.2
1020-P	1250 ± 28	1682 ± 4	6.4 ± 0.4	10.2 ± 0.2	68.2 ± 6.3
1040-P	1258 ± 25	1739 ± 40	7.0 ± 0.6	11.5 ± 0.7	77.2 ± 5.3
1060-P	1237 ± 19	1740 ± 8	6.7 ± 0.3	10.3 ± 0.7	94.6 ± 8.0
1060-NP	1501 ± 34	1820 ± 12	5.7 ± 0.1	7.1 ± 0.3	87.4 ± 6.2

YS, UTS, UE, TE and IT stand for yield strength, ultimate tensile strength, uniform elongation, total elongation and impact toughness, respectively.

## Data Availability

The data that support the findings of this study are available from the corresponding author upon reasonable request. Source data are provided with this paper.

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
