# Peer review of "Effects of Austenitizing Temperature on Tensile and Impact Properties of a Martensitic Stainless Steel Containing Metastable Retained Austenite"

_materials, 2021, doi:10.3390/ma14041000_

Round 1
Reviewer 1 Report
Dear authors,
thank you very much for submitting an interesting paper dealing with the influence of the austenitization temperature on the tensile strength and impact toughness of martensitic stainless steel with the content of retained austenite.
You reached a lot of interesting results. Before, accepting the paper, I recommend doing the following major revisions:
Line 11: You wrote about the effect of austenitizing temperature. Provide also the information on all temperatures used within the study.
Line 50: In the present work, the microstructure and mechanical properties of a MCMSS has been studied, after subjected to different austenitizing temperatures and identical Q&P processes.
Please, check the English. Term „after subjected“ is not correct.
Line 55: By revealing the microstructure and mechanical properties of MCMSSs along with comparisons to experiments data from different austenitizing temperature, the intrinsic link of process-micro-structure-properties has been discussed in detail, aiming for distinguish which and how the decisive factor on ductility and impact toughness of the MCMSSs.
Please, check the English. Some word is missing at the end of the whole sentence.
Line 101: Please, also add the information on etching time.
Line 113: Please, add the time of electro-polishing.
Line 134: What is the M23C6 content at the mentioned temperature of Ac3? Please, check your typo, you wrote Ae3.
Line 135: What about the content of MC at the temperature of 974 °C?
Line 148: How was the carbon diffusion distance in austenite caused by auto-tempering measured?
Line 164: Therefore, according to the alloying elements distribution maps, the reduction of solute chromium and carbon in prior austenite is the main reason for the increase of Ms in 1000-Q sample.
Increase of Ms in 1000-Q sample? What increase, give the value, please. How was Ms calculated?
Line 168: Please, enlarge the arrows in Fig. 3 to better see them.
Line 175: What about the influence of the austenitization temperature on the M23C6 content?
Line 204: Reference is missing for the Equation 3.
Line 220: Please, add some discussion to Fig. 6. Why could be austenite content higher for partitioned samples higher at the temperatures above 1020 °C?
Line 239: Red circles in Figure 8 c,d are not visible.
Line 252: Reference is missing for Equation 4.
Line 306: Equation for Diffusion coefficient D could be written in the new line.
Line 311: This indicates the carbon content in sub-micron-size blocky RA is not homogenization…
Please, check the English.
Line 313: What about the size of RA grains for 1040-P and 1060-P?
Line 465: The effect of austenitizing tempering on microstructure of MCMSSs have been systematically studied and discussed in terms of tensile and impact properties, the following conclusions are achieved.
Write rather: The effect of austenitizing temperature…
Reviewer 2 Report
On Fig,6a the Authors show the dependence between austenitizing temperature and C content in retained austenite. The C content value in retained austenite for sample 1060-NP is unusual.
The Authors did not conduct a detailed analysis for this result. Such high C content in retained austenite for austenitizing temperature equal 1600 Ëš is rather impossible to achieve. The test results presented in Fig. 6b should be supplemented with an analysis of this case.
Line 429 - mistake
The Authors write about large carbides, but put the symbol of nitrides (TiN) – please correct.
Reviewer 3 Report
The authors carry out a research work on medium carbon martensitic stainless steels (0.23% C and 12.5% Cr), whose objective is to relate the retained austenite with various austenitizing temperatures and analyze the variation of mechanical properties with the variation of said austenitizing temperature.
It is important to highlight that the quenching of the samples was carried out in oil and that the tempering temperatures were 250 ºC. This is a good paper, and I congratulate the authors for the effort made to present clear and didactic figures. I only suggest some minor modifications to the manuscript.
I recommend expanding the introduction to include comments and references on:
- the influence of the temperatures Ms and Mf on the percentage of retained austenite. It should be noted that in this type of steels, the temperature Ms is quite low and that normally the ambient temperature is between the temperatures Ms and Mf.
- the influence of quenching in oil or in air.
- the influence of the tempering temperature. There are two tempering options for these steels: Tempering temperature at 500ºC or 200-250ºC. Explain how these two temperatures can influence on the percentage of retained austenite. (500 ºC may favor a second destabilization of the austenite).
In the results I suggest expanding the information in the caption of Figure 4.
Reviewer 4 Report
The current paper investigates the impact of austenization temperature on the microstructure and mechanical characteristics of stainless steel type of material using non destructive techniques, impact toughens and tensile test. The results reported by authors indicate improved mechanical properties and enhanced ductility.
The article is interesting. Authors should carefully study the comments and make improvements to the article step by step. After minor changes the article can be considered for publication in the "Materials" Journal.
Literature review is very basic, it does not reflect on previous studies and what has been done in the past, the authors must prepare a concise but comprehensive literature review on some of the previous work done on same or similar type of steel.
The authors are encouraged to add some images in section 2 to show the equipment and test setup
Line 225 what do the authors mean by the word block
Figure 8 e to h are not clear either remove them or replace them with more clear images
Figure 9 for 1000-P there is a great variation compared to other samples, did the authors attempt to make additional tests to check whether this has to do with sample quality or test errors?
Round 2
Reviewer 1 Report
Dear authors,
thank you very much for providing the revised version of your manuscript. Currently, I recommend accepting your paper for publication.
Best regards
Reviewer